# Colistin Use in Patients with Chronic Kidney Disease: Are We Underdosing Patients?

**DOI:** 10.3390/molecules24030530

**Published:** 2019-02-01

**Authors:** Luisa Sorli, Sonia Luque, Jian Li, Eva Rodríguez, Nuria Campillo, Xenia Fernandez, Jade Soldado, Ignacio Domingo, Milagro Montero, Santiago Grau, Juan P. Horcajada

**Affiliations:** 1Infectious Diseases Department, Hospital del Mar, Infectious Pathology and Antimicrobial Research Group (IPAR), Institut Hospital del Mar d’Investigacions Mèdiques (IMIM), Universitat Autònoma de Barcelona (UAB), CEXS-Universitat Pompeu Fabra, 08003 Barcelona, Spain; 61762@parcdesalutmar.cat (J.S.); 62887@hospitaldelmar.cat (I.D.); 95422@parcdesalutmar.cat (M.M.); JHorcajada@hospitaldelmar.cat (J.P.H.); 2Spanish Network for Research in Infectious Diseases (REIPI RD 16/0016/0015), Instituto de Salud Carlos III, 28001 Madrid, Spain; FAR1207@parcdesalutmar.cat (N.C.); Xfernandez@parcdesalutmar.cat (X.F.); SGrau@hospitaldelmar.cat (S.G.); 3Pharmacy Department, Hospital del Mar, Infectious Pathology and Antimicrobial Research Group (IPAR), Institut Hospital del Mar d’Investigacions Mèdiques (IMIM), Universitat Autònoma de Barcelona (UAB), CEXS-Universitat Pompeu Fabra, 08003 Barcelona, Spain; 4Monash Biomedicine Discovery Institute, Department of Microbiology, Monash University, Clayton, 3800 Victoria, Australia; Jian.Li@monash.edu; 5Nephrology Department, Hospital del Mar, Institut Hospital del Mar d’Investigacions Mèdiques (IMIM), Universitat Autònoma de Barcelona (UAB). CEXS-Universitat Pompeu Fabra, 08003, 08003 Barcelona, Spain; ERodriguezG@parcdesalutmar.cat

**Keywords:** colistin, colistin plasma concentrations, chronic kidney disease, pharmacokinetic, toxicodynamic

## Abstract

Colistin is administered as its inactive prodrug colistimethate (CMS). Selection of an individualized CMS dose for each patient is difficult due to its narrow therapeutic window, especially in patients with chronic kidney disease (CKD). Our aim was to analyze CMS use in patients with CKD. Secondary objectives were to assess the safety and efficacy of CMS in this special population. In this prospective observational cohort study of CMS-treated CKD patients, CKD was defined as the presence of a glomerular filtration rate (GFR) < 60 mL/min/m^2^ for more than 3 months. The administered doses of CMS were compared with those recently published in the literature. Worsened CKD at the end of treatment (EOT) was evaluated with the RIFLE (Risk, Injury, Failure, Loss of kidney function, and End-stage kidney disease) criteria. Colistin plasma concentrations (C_ss_) were measured using high-performance liquid chromatography. Fifty-nine patients were included. Thirty-six (61.2%) were male. The median age was 76 (45–95) years and baseline GFR was 36.6 ± 13.6. The daily mean CMS dosage used was compared with recently recommended doses (3.36 vs. 6.07; *p* < 0.001). Mean C_ss_ was 0.9 (0.2–2.9) mg/L, and C_ss_ was <2 mg/L in 50 patients (83.3%). Clinical cure was achieved in 43 (72.9%) patients. Worsened renal function at EOT was present in 20 (33.9%) patients and was reversible in 10 (52.6%). The CMS dosages used in this cohort were almost half those currently recommended. The mean achieved C_ss_ were under the recommended target of 2 mg/dL. Despite this, clinical cure rate was high. In this patient cohort, the incidence of nephrotoxicity was similar to those found in other recent studies performed in the general population and was reversible in 52.6%. These results suggest that CMS is safe and effective in patients with CKD and may encourage physicians to adjust dosage regimens to recent recommendations in order to optimize CMS treatments.

## 1. Introduction

Colistin has emerged as a last resort drug for the treatment of infections caused by multidrug and extensively drug-resistant Gram-negative bacteria such as *Pseudomonas aeruginosa* and *Acinetobacter baumannii* [1,2]. Although several clinical studies have evaluated the pharmacokinetics (PK) of colistin and its prodrug colistimethate sodium (CMS) in different types of patients [3,4,5,6,7,8,9,10] in the last decade, there is still little information on both over and underdosing in high-risk clinical settings, such as chronic kidney disease (CKD) and acute kidney injury (AKI) [4,5,11,12].

Colistin is a concentration-dependent bactericidal antibiotic with a narrow therapeutic window, with nephrotoxicity being its major dose limiting adverse effect [13,14,15]. Until recently, the CMS doses used in daily clinical practice were based on product manufacturer information with limited and rudimentary pharmacological information [2,11]. However, population pharmacokinetic (PK) studies performed in the last few years have led to new dosage schedule recommendations for both patients with normal and reduced renal function, including those undergoing renal replacement therapies [8,11,12]. These new guidelines recommend CMS dose selection and adjustment based on baseline estimated glomerular filtration rate (GFR) or creatinine clearance, as well as on the target colistin plasma concentration (C_ss_). This value, based on different data from in vitro and in vivo PK, pharmacodynamic (PD) and toxicodynamic (TD) studies has been defined as 2 mg/L [16]. All this scientific research has led to the publication of the first consensus document [17]. However, a recent publication has suggested that current recommendations on the use of colistin in patients with reduced renal function are likely to be inadequate [11] and have updated the dosing guidelines of both the European and American regulatory agencies [18,19]. 

The aim of this study was to evaluate colistin use in a cohort of patients with CKD and to assess whether “old recommendations” are suitable from a PK/PD/TD point of view. A secondary objective was to evaluate the potential factors involved in clinical failure and worsening renal function (WRF) at the end of treatment (EOT). 

## 2. Results

During the study period, 59 patients were enrolled, including 36 (61%) men with a median age of 76 years (interquartile range [IQR], 45–95). All of them had infections caused by extensively drug-resistant *Pseudomonas aeruginosa:* pneumonia in 14 (23.7%), acute bronchitis in 14 (23.7%), urinary tract infection in 16 (27.1%), bacteremia in three (5.1%), skin and soft tissue infections in five (8.6%) and miscellanea in seven (11.9). The mean estimated GFR at baseline was 36.6 ± 13.6 SD and the median CMS daily dose was 3 (IQR, 1 to 9) million international units (IU). A loading dose was administered in six (10.17%) patients, with the median dose being 6 million IU (IQR, 3–6). Among patients receiving a loading dose, two were diagnosed with a UTI, one with pneumonia and three with tracheobronchitis. None of the included patients was undergoing renal replacement therapy during the study period. WRF was present in 20 (33.9%) patients at EOT. Thirty-day all-cause mortality was 28.8%. Patient characteristics and CMS concentrations are shown in Table 1. The median C_ss_ was 0.9 mg/L, but individual values varied widely (IQR 0.2–2.9 mg/L) with the physician–selected doses. The distribution of C_ss_ in different GFR clusters is shown in Figure 1. 

The median (IQR) of C_ss,avg_ were 0.9 (0.2–2.9), 0.83 (0.4–1.6), 1.2 (0.2–2.9), 0.9 (0.28–1.4) and 0.7 (0.4–1.1) mg/L for the GFR clusters of 50–60, 40–50, 30–40, 20–30 and 10–20 mL/min/1.73m^2^, respectively. In 50 (83.3%) patients the concentration of formed colistin in plasma was below the suggested therapeutic level of 2 mg/L [12].

Regarding the received CMS doses, 41 (68.3%) patients received daily doses <3 million IU per day, 16 (26.7%) between 3–6 million IU per day and only three (5%) received more than 6 million IU per day. The median (IQR) of C_ss,avg_ in these three patient groups was 0.8 (0.2–2.23), 1.25 (0.3–2.9) and 0.68 (0.5–0.86) mg/L, respectively. Table 2 shows the medians of CMS and CBA suggested by Nation et al. [12] for a desired target of C_ss,avg_ of 2 mg/L compared with those administered in this cohort of patients. 

The mean dose received by patients in our study was significantly lower than those suggested by the study of Nation et al. [12] (3.36 ± 0.23 vs. 6.07 ± 0.11; *p* < 0.001). Because the recommendations for colistin doses in our center changed in 2014, we analyzed the doses used in the first period (from 2010 to December 2014) and in a second period from 2014 to the present (2014–2018) and there were no changes between these two periods (mean dose 3.15 ± 1.64 vs. 3.57 ± 1.93; *p* = 0.37).

Clinical cure was achieved in 43 (72.8%) patients. Patients with clinical failure were more severely ill (SOFA 6 vs. SOFA 2; *p* < 0.001), had achieved higher colistin plasma concentrations (0.8 vs. 1.1, *p* = 0.03), and had higher rates of 30-day all-cause mortality (81.3% vs. 9.3%); *p* < 0.001). The characteristics of patients with and without clinical cure are shown in Table 3. On multivariate analysis, the only factor related to clinical failure was SOFA score (OR 0.63; 95% CI, 0.44–0.90; *p* = 0.012). The overall incidence of WRF at the EOT was 33.9%. The median time from colistin initiation to WRF was 7 (IQR 1 to 16) days. WRF was reversible in 10 (52.6%) patients, but follow up results were lacking in six (31.6%). Patients with nephrotoxicity at EOT had higher median (IQR) SOFA scores (3 (1–9) versus 2 (0–9); *p* = 0.16), had a longer CMS treatment (15.5 days (6–30) versus 12 days (4–45); *p* = 0.12), had received more concomitant nephrotoxic drugs (1.6 ± 1 versus 1.2 ± 0.9; *p* = 0.15) and achieved higher C_ss,avg_ (1.2 ( 0.3–2.9) versus 0.83 ( 0.2–2.4); *p*=0.1) mg/L. Characteristics of patients with and without WRF at EOT are shown in Table 4. In the multivariate analysis, the only factors related to nephrotoxicity at EOT were C_ss_ (OR 3.21; 95% CI 1.02–10.1; *p* = 0.047) and days of CMS treatment (OR 1.11; 95% CI 0.99–12.4; *p* = 0.069). The results of this analysis are shown in Table 5. Of importance, none of the patients in this cohort developed neurotoxicity.

## 3. Discussion

Colistin use has reemerged in recent years for the treatment of multidrug-resistant Gram-negative infections [1,2,20], which has prompted the performance of a large number of clinical and PK studies of its use in the last decade. However, there is still little pharmacological information in patients at high risk of under- or overdosing, such as those with CKD. Colistin is administered in the inactive form of CMS, which is hydrolyzed spontaneously in vivo to colistin, the active compound. CMS, the pro-drug, is renally eliminated by glomerular filtration and active tubular secretion. Previous experiences have reported that about 60% of the CMS dose is renally excreted in the urine during the first 24 h after dosing [21]. Similarly, a study performed by our group confirmed that there is a rapid urinary excretion of CMS in patients within the first 6 h after intravenous administration [22]. In patients with renal dysfunction, urinary CMS excretion is reduced and, consequently, a larger fraction of this pro-drug can be converted to colistin, leading to drug overexposure in this population [8,21] and consequently to a high risk of kidney injury. Selection of the CMS dose in patients with CKD is therefore a critical issue [13,23,24,25].

Recent toxicodynamic studies performed in the general population indicate that the risk of nephrotoxicity in patients receiving CMS increases with plasma colistin concentration exceeding 2.5 mg/L [13,15]. This point, together with the lack of information on the minimum inhibitory concentrations (MIC) at the beginning of CMS treatment, have led to consideration of an average C_ss,avg_ of 2 mg/L as a target when initiating therapy [11]. The median C_ss,avg_ in this study was 0.9 mg/L, which is far below the above-mentioned recommended 2 mg/L [11] but is also below those reported in previous studies performed by our group in the general population (1.06 with an IQR of 0.11–5.99) [7,13]. This finding is surprising, since patients with a reduced GFR could have been expected to achieve higher colistin plasma concentrations [8,11,12]. An explanation could be the low CMS doses administered in our patient cohort because of the use of outdated product/hospital CMS dosing recommendations, as well as clinicians’ concern about nephrotoxicity in patients with compromised renal function at baseline. Another important finding of this study is the considerable interpatient variability in colistin plasma concentrations even in patients within the same GFR range (Figure 1). This finding has also been reported in recent pharmacokinetic studies. Nation et al. reported up to ~12-fold inter–patient variability in plasma colistin C_ss_ across all four renal function groups (≥80, 50 –<80, 30–<50 and <30 mL/min) [11]. A previous study by our group also reported wide variability in observed plasma colistin concentrations (1.06 with an IQR of 0.11 to 5.99) [13]. In addition, in a pharmacokinetic study, Garonzik et al. found wide interindividual variability with a range of the colistin C_ss,avg_ of 0.48–9.38 mg/L (median, 2.36 mg/L) [8].

Until a few years ago, CMS dose selection was guided by the recommendations in the product information sheet, based on outdated pharmacological information [2] and in local protocols. As of 2013, new PK, PD and clinical knowledge led to the new dosing guidelines [17], which has been published by several regulatory agencies such as the European Medicines Agency (EMA) [19] and the US Food and Drug Administration (FDA) [18]. However, their impact and implementation in daily clinical practice are still unknown. Additionally, a recent publication highlights substantial differences between the FDA- and EMA-approved CMS dose recommendations and suggests that these doses are inadequate to achieve the PK target of 2 mg/L in certain clinical settings [11]. In this scenario, based on a population pharmacokinetic study [8], Nation et al. proposed a clinician-friendly dosing algorithm [12]. When we compared the CMS doses proposed in this algorithm with the real doses used in our daily clinical practice, we observed that patients in the present cohort were underdosed (3.36 ± 0.23 vs. 6.07 ± 0.11; *p* < 0.001)., which could explain the low C_ss_ found in the patient cohort in our hospital.

The overall rate of clinical cure in this patient cohort was 72.8%, which is similar to rates reported by other studies [7,26,27,28]. The only studied factor associated with clinical failure was disease severity (SOFA). Of note, colistin plasma levels were not related to clinical cure. This finding was also observed in our previous study [7], and was somewhat in disagreement with recent pharmacokinetic studies suggesting that a C_ss_ of 2 mg/L could be a desirable target for the treatment of infections caused by MDR-GNB [8,11,12]. However, the rates of clinical cure differed, depending on the patients’ diagnosis, and this percentage was only 50% in patients diagnosed with pneumonia. This finding confirms the hypothesis of Nation et al. that intravenous CMS is not efficacious against lung infections [16] and therefore a plasma colistin concentration of 2 mg/L may not be adequate for isolates with a colistin MIC >1 mg/L [11]. In this scenario, higher CMS doses to achieve adequate PK/PD targets could lead to the development of nephrotoxicity, since previous studies have identified colistin plasma concentrations >2.5 mg/L as a risk factor for nephrotoxicity [13,14,15].

Although the small sample size of this study does not allow definite conclusions to be drawn, we advocate treatment of lung infections with inhalation or combined therapy, as well as application of therapeutic drug monitoring to optimize PK/PD while minimizing the risk of nephrotoxicity. In contrast, in less severe infections such as UTI, a plasma level lower than the proposed 2 mg/L might be enough. Consequently, we believe that it is difficult to define a single optimal C_ss_ level for all types of infections and microorganisms and that it would be better to have a well-defined PK/PD ratio as a target.

The rate of WRF in this cohort of patients with CKD was 33.9% and nephrotoxicity is the main adverse effect of polymyxins. Rates of nephrotoxicity in patients treated with CMS range from 20% to 60% according to recent studies using standardized criteria to evaluate nephrotoxicity [7,13,23,24,25]. Even though WRF was reversible in more than 50% of patients, the diagnosis and management of this complication during CMS treatment is a challenge in daily clinical practice, since nephrotoxicity is potentially related to worse clinical outcomes. In two previous studies by our group, the presence of AKI at EOT was a predictor of 30-day all-cause mortality [7,13]. In another study performed by Falagas et al. in 258 patients with infections caused by multidrug-resistant Gram-negative bacilli (GNB), the development of AKI during treatment was also related to mortality [20]. Several studies have assessed the risk factors for colistin-associated nephrotoxicity with different results [23,24,26,29,30], but recent clinical PK/TD studies have indicated that the risk of nephrotoxicity increased as plasma colistin exposure exceeded approximately 2.5 mg/L [13,14,15]. Indeed, the present study also demonstrates that C_ss_ is a predictor of WRF at EOT.

In conclusion, intravenous CMS use in patients with CKD with the currently recommended dosage regimens is just as safe as that in the general population. Our data suggest that the dosage regimens used in daily clinical practice should be updated based on modern PK/PD/TD studies, at least in the treatment of MDR infections. In this scenario, the use of the formula of Nation et al. [12] or possibly the clinician-friendly dosing algorithm proposed by the same group [12] could be a useful clinical tool to guide treatment in this special population. Another important conclusion of this study is that rates of clinical cure in patients with pneumonia are poor with the dosage regimens used in daily clinical practice. Regarding colistin-associated nephrotoxicity, this study confirms the relationship between colistin plasma concentrations and the development of nephrotoxicity during treatment. Finally, all these issues highlight the need for therapeutic drug monitoring in daily clinical practice both to optimize PK/PD and to prevent the development of nephrotoxicity.

## 4. Materials and Methods

### 4.1. Study Population

We conducted a prospective observational study in a cohort of patients with CKD and infections caused by extensively drug-resistant *P. aeruginosa* treated with intravenous CMS for at least 48 h. The study was carried out at Hospital del Mar, a tertiary care university hospital in Barcelona (Spain), from January 2010 to August 2018. The study was approved by the local ethics committee (Comité Ètic d’Investigació Clínica del Parc de Salut Mar. Approval number 2011/4501/I). Exclusion criteria were age <18 years, pregnancy, and breast-feeding during the study period. A pharmacy-generated alarm system was used to identify patients under CMS treatment. The study investigators performed the assessment from the first day of the treatment and informed consent was obtained from all participants or their legal representatives.

### 4.2. Collected Data

Patient data included demographic information, SOFA score [31], Charlson comorbidity index [32], CMS treatment (indication, daily and total cumulative dose measured in millions of IU and treatment duration). GFR at baseline and at EOT was calculated using the abbreviated Modification of Diet in Renal Disease equation (MDRD-4) [33]. WRF was defined as a decrease in GFR ≥25% and was classified according to the Risk, Injury, Failure, Loss of kidney function, and End-stage kidney disease (RIFLE) criteria [34]. Patients with WRF were followed up until GFR recovery or until hospital discharge. Information was collected on the concomitant use of other potential nephrotoxic drugs such as aminoglycosides, vancomycin, angiotensin II receptor blockers, angiotensin-converting enzyme (ACE) inhibitors, loop diuretics, intravenous dye, amphotericin B and non-steroidal anti-inflammatory drugs (NSAIDs), as well as the need for vasopressor drugs or discontinuation of CMS due to nephrotoxicity. Combination therapy with other antibiotics with potential activity or synergy with colistin was assessed.

### 4.3. Definitions

CKD was defined as kidney damage or GFR <60 mL/min/1.73 m^2^ for 3 months or more, irrespective of cause [35]. Infections were defined according to the Centers for Disease Control and Prevention [36]. Clinical failure was defined as a lack of improvement in patients or in at least one of the initial symptoms, worsening or death. Clinical cure was defined as either the absence of symptoms or as a consistent improvement in the signs and symptoms of the infection. Thirty-day all-cause mortality was considered as death from any cause during the 30 days following EOT. Combined antibiotic treatment consisted of CMS plus meropenem or CMS plus amikacin.

### 4.4. CMS Administration

The dosage regimen and dose adjustments were determined by the treating physician. Since this study was performed over more than 8 years, colistin daily doses varied during this time. From 2010 to 2014, dose adjustments were made according to the package insert’s recommended dosing as follows: GFR ≥76 mL/min/1.73 m^2^, 4–6 million IU daily in three doses; GFR 40–75 mL/min/1.73 m^2^, 2–3 million IU daily in two doses; GFR 25–40 mL/min/1.73 m^2^, 1.5–2 million IU daily divided in one or two doses and; GFR <25 mL/min/1.73 m^2^, 0.6–1 million IU daily every 36 h. In 2014, a new protocol based on our previous experience [13] was implemented in our center and CMS was adjusted as follows: GFR ≥90 mL/min/1.73 m^2^, 9 million IU daily in three doses; GFR 50–89 mL/min/1.73 m^2^, 3 million IU every 12 h, GFR 10–49 mL/min/1.73 m^2^, 3 million IU per day; GFR ≤10 mL/min/1.73 m^2^, 2 million IU/day. However, the final dose was chosen by the treating physician. CMS (colistimethate formulation for intravenous use, GES Genéricos Españoles^®^, Las Rozas, Spain was diluted in 100 mL of physiological saline before intravenous administration over 30 min. Each vial contained 1 million IU of CMS (equivalent to 80 mg CMS).

### 4.5. Microbiological Data

Identification and susceptibility testing of *P. aeruginosa* were first performed by microdilution using the Gram-negative breakpoint panel for non-fermenting GNB of the MicroScan^®^ WalkAway system (Siemens Diagnostic Inc., Los Angeles CA, USA). Colistin MIC was determined by microdilution using cation-adjusted MHB; the isolate was considered susceptible if the MIC was ≤ 2 mg/L according to the Clinical and Laboratory Standards Institute [37].

### 4.6. Pharmacokinetic Data

Blood samples were obtained just before the next dose on day 3–4 of the treatment. It was assumed that steady state was already achieved considering a half-life of approximately 14 h [10]. Concentrations of CMS and formed colistin in plasma were measured using a validated high-performance liquid chromatography (HPLC) method [13,38]. The limit of quantification of the HPLC methods for colistin and CMS in plasma were 0.20 and 0.50 mg/L, respectively. As the plasma concentration-time profiles of formed colistin are almost flat [8,13], measured colistin concentrations were regarded as C_ss,avg_. The percentage of patients achieving the therapeutic target defined as a C_ss,avg_ of 2 mg/L was also analyzed. Finally, the CMS doses used in our cohort of patients in daily clinical practice were compared with those recently recommended by the algorithm of Nation et al. [12]. We calculated the expected C_ss_ based on the formula of Nation et al. [12].

### 4.7. Statistical Analysis

Dichotomous data were compared using a χ^2^ or Fisher’s exact test. Normally distributed continuous data are expressed as means and standard deviations (SD), and were compared using the *t*-test. Otherwise, values are presented as means with interquartile range (IQR) and were compared using the Mann-Whitney U-test. The baseline and clinical characteristics of patients receiving different CMS doses were compared using the ANOVA or Kruskal-Wallis test. Multivariate analysis of risk factors for colistin-associated nephrotoxicity was conducted using logistic regression. Univariate analyses were performed separately for each of the risk factor variables to ascertain the odds ratio (OR) and 95% confidence interval (CI). All clinically important covariates and those with *p* < 0.3 in the univariate analyses were included in the multivariate analysis.

## Figures and Tables

**Figure 1 molecules-24-00530-f001:**
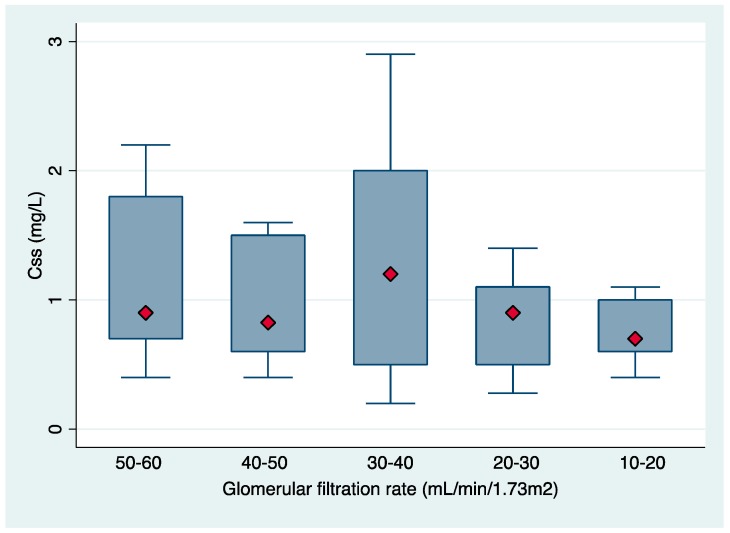
Boxplot of achieved C_ss,avg_ in different intervals of baseline glomerular filtration rate.

**Table 1 molecules-24-00530-t001:** Patients’ clinical, demographic and pharmacokinetic characteristics.

Variable	Patients (*n* = 59)
Age, years *	76 (45–95)
Male sex, n (%)	36 (61)
SOFA *	2 (0–9)
Charlson score **	6.86 ± 2.26
BMI (Kg/m^2^) *	26.5 (14.7–52.1)
Intensive care unit admission, n (%)	11 (18.6)
Type of infection:	
Pneumonia	14 (23.7)
Acute bronchitis	14 (23.7)
Urinary tract infection	16 (27.1)
Bacteremia	3 (5.1)
Skin and soft tissue infection	5 (8.6)
Others	7 (11.9)
GFR at baseline (mL/min/1.73 m^2^) **	36.6 ± 13.6
GFR intervals((mL/min/1.73 m^2^):	
50 to <60	9 (15.3)
40 to <50	14 (23.7)
30 to <40	18 (30.5)
20 to <30	11 (18.5)
10 to <20	7 (11.9)
CMS daily dose (million IU) *	3 (1–9)
Recommended daily dose (million IU) **	6.1 ± 0.81
CMS daily dose (mg)/kg *	3.33 (1.12–10.7)
CBA (mg)/kg *	1.39 (0.47–4.44)
C_ss,avg_ (mg/L) *	0.9 (0.2–2.9)
Expected C_ss,avg_ (mg/L) *	0.98 (0.37–2.46)
CMS total dose (million IU)	40.5 (6–335)
CMS duration, days *	14 (4–75)
GFR at EOT (mL/min/1.73 m^2^) *	30.1 (6.51–81.4)
WRF at the end of treatment, n	20 (33.9)
R (Risk)	3 (15)
I (Injury)	10 (50)
F (Failure)	3 (15)
Recovery of baseline renal function	10 (52.6)
Combined antibiotic treatment	32 (54.2)
30-day all-cause mortality	17 (28.8)

* Median (interquartile range). ** Mean ± standard deviation. Abbreviation notes: BMI: body mass index; GFR: glomerular filtration rate; CBA: colistin base activity; CMS: colistin menthanesulfonate; C_ss,avg_: colistin plasma concentration at steady-state; EOT: end of treatment; SOFA: sequential organ failure assessment score; WRF: worsened renal function.

**Table 2 molecules-24-00530-t002:** Table daily doses of colistimethate for a desired target colistin C_ss_ of 2 mg/L compared with the mean daily doses used in daily clinical practice for different creatinine clearances.

Creatinine Clearance mL/min	Dose of Colistimethate for C_ss,avg_ of 2 mg/L *		Dose of Colistimethate in Daily Clinical Practice	
	CBA, mg/d	Million IU/d	CBA, mg/d	Million IU/d
50 to <60	245	7.4	174	5.22
40 to <50	220	6.65	115	3.46
30 to <40	195	5.9	116	3.47
20 to <30	175	5.3	84.4	2.59
10 to <20	160	4.85	57.1	1.71

* Nation et al. CID 2017. Abbreviations notes: CBA: colistin base activity; C_ss_: colistin plasma concentration.

**Table 3 molecules-24-00530-t003:** Univariate analysis of patients with and without clinical cure.

Variable	Patients with Clinical Cure (*n* = 43)	Patients with Clinical Failure (*n* = 16)	*p*
Age (years)	76 (45–95)	76 (52–91)	1
Male, sex	24 (55.8)	12 (75)	0.25
SOFA	2 (0–6)	6 (1–9)	<0.001
Charlson score	7.21 ± 2.27	7.94 ± 1.53	0.23
BMI (kg/m^2^)	26.7 (20.9–52.1)	24.8 (14.7–41.6)	0.12
Diagnosis:			0.05
Pneumonia	7 (50)	7 (50)	
UTI	15 (93.6)	1 (6.25)	
Tracheobronchitis	11 (78.6)	3 (21.43)	
SSTI	3 (60)	2 (40)	
Bacteremia	3 (100)	0 (0)	
Others	4 (57.1)	3 (42.9)	
CMS daily dose (IU)	3.3 ± 1.7	3.5 ± 2.05	0.66
CBA daily dose (mg)	110.1 ± 57.1	117.7 ± 68.2	0.66
CMS daily dose (IU/kg)	3.75 ± 2.10	4.25 ± 2.69	0.45
CBA daily dose (mg/kg)	1.56 ± 0.88	1.77 ± 1.12	0.45
GFR at baseline (mg/kg/m^2^)	36.6 (11.1–70.1)	38.5 (19.9–59.1)	0.39
CMS total dose (IU)	40 (7–167)	31.6 (18–148.5)	0.8
CMS duration treatment (days)	14 (4–45)	12 (5–30)	0.6
C_ss,avg_	0.8 (0.2–2.4)	1.1 (0.3–2.9)	0.03
Expected C_ss,avg_	0.98 (0.38–2.46)	0.98 (0.37–2.34)	0.78
Nephrotoxicity at the EOT	12 (27.9)	8 (50)	0.13
30-day all-cause mortality	4 (9.3)	13 (81.3)	<0.001

Abbreviation notes: BMI: body mass index; CBA: colistin base activity; CMS: colistin menthanesulfonate; C_ss,avg_: colistin plasma concentration at steady-state; EOT: end of treatment; GFR glomerular filtration rate; SOFA: sequential organ failure assessment score; SSTI: skin and soft tissue infection; UTI: urinary tract infection.

**Table 4 molecules-24-00530-t004:** Characteristics of patients with and without acute kidney injury at the end of treatment.

Variable	Patients with Nephrotoxicity (*n* = 20)	Patients without Nephrotoxicity (*n* = 39)	*p*
	End of Treatment		
Age, years	76 (57–91)	76 (46–95)	0.97
Male, sex	14 (70)	22 (56.4)	0.3
Weight	72.4 ± 17.4	72.4 ± 13.9	1
BMI	24.5 (15.9–52)	26.7 (14.7–41.6)	0.5
SOFA	3 (1–9)	2 (0–9)	0.16
Charlson score	7.4 ± 2.2	7.42 ± 2.10	0.97
Intensive care unit admission	6 (30)	5 (12.8)	0.1
BMI (kg/m^2^)	24.48 (15.94–52.1)	26.74 (14.69–41.64)	0.19
CMS daily dose (IU)	3 (1–6)	3 (1–9)	0.45
CBA daily dose (mg)	100 (33.3–200)	100 (33.3–300)	0.5
CMS daily dose (IU/kg)	3 (1.22–10.7)	3.43 (1.12–0.14)	0.8
CBA daily dose (mg/kg)	1.25 (0.51–4.44)	1.43 (0.47–3.8)	0.8
GFR at baseline (mg/kg/m^2^)	36.2 ± 9.6	36.7 ± 15.3	0.9
Suggested dose (IU)	6 ± 0.64	6.1 ± 0.9	0.9
CMS total dose (IU)	39.5 (16–125)	36 (7–167)	0.9
CMS duration treatment (days)	15.5 (6–30)	12 (4–45)	0.12
C_ss_	1.2 (0.3–2.9)	0.83 (0.2–2.4)	0.1
Expected C_ss_	0.98 (0.39–2.09)	0.98 (0.37–2.46)	0.5
Concomitant nephrotoxic drugs:	18 (90)	30 (76.9)	0.3
Aminoglycosides	1 (5.55)	5 (16.7)	0.014
Loop diuretics	16 (88.9)	22 (73.3)	0.09
NAIDS	3 (16.7)	8 (26.7)	0.7
ACE inhibitors	4 (22.2)	8 (26.7)	1
Intravenous dye	2 (11.1)	5 (16.7)	1
Number of nephrotoxic drugs	1.6 ± 0.99	1.2 ± 0.98	0.15
30-day all-cause mortality	7 (35)	10 (25.64)	0.45

Abbreviation notes: ACE: angiotensin-converting enzyme; BMI: body mass index; CBA: colistin base activity, CMS colistin menthanesulfonate; GFR: glomerular filtration rate; C_ss_: colistin plasma concentration; NSAIDS: non-steroidal anti-inflammatory agents; SOFA: sequential organ failure assessment score.

**Table 5 molecules-24-00530-t005:** Multivariate analysis of independent risk factors for colistin-associated nephrotoxicity at the end of treatment.

	OR	95% CI	*p* Value
DOT	1.11	0.99–12.4	0.069
C_ss_	3.21	1.02–10.1	0.047

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
