# Peer review of "Colistin Use in Patients with Chronic Kidney Disease: Are We Underdosing Patients?"

_molecules, 2019, doi:10.3390/molecules24030530_

Round 1
Reviewer 1 Report
The presentation of the data is insufficient, several questions remain open:
Was a starting dose given? Usually on the first day a higher dose is administered because especially in pneumonia the first 24h are crucial and sufficient plasma levels are needed from the start.
No distinction is made between patients with and without continuous hemofiltration, although several patients had GFR´s < 15ml/min. Doses have to be adjusted for hemofiltration
Data on other nephrotoxic drugs were collected but not presented - did patients with WRF receive NSAR´s with the colistin etc.
Did patients with signs of nephrotoxicity also show neurotoxicity?
The most important question: Why were the doses so low?
For comparison the dosing regime at University Hospital Heidelberg:
day 1: - one single dose of 9 Mio. IU (in criticall ill patients up to 12 Mio. IU).
from day 2 (24 hours after starting dose):
- GFR 30-60 ml/min: 4 Mio. IU every 12 hours
- GFR 15-<30 ml/min: 3 Mio. IU every 12 h
- GFR<15 ml/min: 2 Mio. IU every 12 h
with continuous hemofiltration (CVVHD with dialysate flow 42ml/min, blood flow 150 ml/min, membrane 0,9 m²): 16 Mio.IU in 2-3 doses /day
These doses are all twice the doses given in the presented patient cohort, but no explanation is offered why such low doses were chosen.
Author Response
The authors would like to thank the Reviewers for their careful review of tor manuscript and for providng us with their coments and suggestions to improve the quality of the manuscript. The following responses have been prepared to address all the reviewer’s comments in a point-by-point fashion.
Point 1: Was a starting dose given? Usually on the first day a higher dose is administered because especially in pneumonia the first 24h are crucial and sufficient plasma levels are needed from the start.
Response 1: According to the reviewer’s suggestion, data regarding loading dose have been included (Results section, page 2, line 74-78).
Point 2: No distinction is made between patients with and without continuous hemofiltration, although several patients had GFR´s < 15ml/min. Doses have to be adjusted for hemofiltration
Response 2: During the study period, none of the included patients was undergoing any renal replacement therapy.
Point 3: Data on other nephrotoxic drugs were collected but not presented - did patients with WRF receive NSAR´s with the colistin etc.
Response 3: As suggested, new data about the number and the type of nephrotoxic drugs have been added to Table 4.
Point 4: Did patients with signs of nephrotoxicity also show neurotoxicity?
Response 4: authors have included this data (page 4, line 119-121)
Point 5: The most important question: Why were the doses so low? For comparison the dosing regime at University Hospital Heidelberg:
day 1: - one single dose of 9 Mio. IU (in criticall ill patients up to 12 Mio. IU).
from day 2 (24 hours after starting dose):
- GFR 30-60 ml/min: 4 Mio. IU every 12 hours
- GFR 15-<30 ml/min: 3 Mio. IU every 12 h
- GFR<15 ml/min: 2 Mio. IU every 12 h
with continuous hemofiltration (CVVHD with dialysate flow 42ml/min, blood flow 150 ml/min, membrane 0,9 m²): 16 Mio.IU in 2-3 doses /day.
These doses are all twice the doses given in the presented patient cohort, but no explanation is offered why such low doses were chosen
Response 5: Authors have included an explanation of the used doses in the Material and Methods section (page 9, line 263-272). The authors want to point out that despite the existence of a protocol, the final doses were chosen by the physicians responsible for the patient.
Reviewer 2 Report
In this manuscript, authors demonstrated that intravenous administration of colistin was effective for treatment against multi-drug resistant Pseudomonas aeruginosa infection, and relatively safe in patients with chronic kidney disease (CKD) compared with general population. Interestingly, median plasma concentration of colistin was not achieved to the suggested therapeutic level of 2 mg/L, although it led to satisfactory cure rate of 72.8%. The subject of study seems to be clinically significant, and the study was well-designed. The reviewer has some comments as following.
Major comments
1. The reviewer wonders if there were any differences in metabolism of the prodrug between patients with CKD and those with normal renal function. Authors determined Css three to four hours after administration. However, if peak drug levels were delayed, Css might actually achieve to sufficient therapeutic level five hours or more later, resulted in good clinical response. Authors should explain whether there are any changes in its pharmacokinetics in patients with renal dysfunction.
2. Authors suggested the safe use of colistin in patients with CKD. However, how clinicians adjusted dosage of CMS according to renal function has to be shown in order to ensure the safety. Authors should describe the algorism of dosage adjustment in predefined study protocol in detail.
3. Authors should create additional (supplementary) table showing the results of multivariate analysis described in Line 102.
4. Based on this study, can authors suggest the therapeutic target Css level of 1mg/L three to four hours after administration is appropriate in patients with CKD? The reviewer seems that this is an impact finding in this study.
Minor comments
1. In Table1, legend is needed to explain abbreviations and median or mean values.
2. In study protocol, were patients with eGFR<10 or ESRD excluded?
In methods, authors should describe the approval number assigned by ethical committee.
3. Did all patients receive combined antibiotic treatment including colistin? If not, percentage of combined treatment should be shown in Table 1.
Author Response
The authors would like to thank the Reviewers for their careful review of tor manuscript and for providng us with their coments and suggestions to improve the quality of the manuscript. The following responses have been prepared to address all the reviewer’s comments in a point-by-point fashion.
Point 1: In this manuscript, authors demonstrated that intravenous administration of colistin was effective for treatment against multi-drug resistant Pseudomonas aeruginosa infection, and relatively safe in patients with chronic kidney disease (CKD) compared with general population. Interestingly, median plasma concentration of colistin was not achieved to the suggested therapeutic level of 2 mg/L, although it led to satisfactory cure rate of 72.8%. The subject of study seems to be clinically significant, and the study was well-designed. The reviewer has some comments as following.
Major comments
1. The reviewer wonders if there were any differences in metabolism of the prodrug between patients with CKD and those with normal renal function. Authors determined Css three to four hours after administration. However, if peak drug levels were delayed, Css might actually achieve to sufficient therapeutic level five hours or more later, resulted in good clinical response. Authors should explain whether there are any changes in its pharmacokinetics in patients with renal dysfunction.
Response 1: According to the reviewer’s suggestion the authors have added some information regarding CMS metabolism and an explanation about the differences between patients with and without CKD (pag 6, line 146-153).
Colistin is administered in the inactive form of CMS, which is hydrolyzed spontaneously in vivo to colistin, the active compound. CMS, the pro-drug, is renally eliminated by glomerular filtration and active tubular secretion. Previous experiences have reported that about 60% of the CMS dose is renally excreted in the urine during the first 24h after dosing (Li J, et al. JAC 2003). In the same way, a study performed by our group confirmed that there is a rapid urinary excretion of CMS in patients within the first 6 h after intravenous administration (Luque S, et al. AAC 2017).
In patients with renal dysfunction, such as those with CRD or AKI, the CMS urinary excretion is reduced and, consequently, a greater fraction of this pro-drug can be converted to colistin.
Regarding the colistin plasma levels, they were obtained on the 3rd-4th day of CMS treatment, when the steady state has presumably had been achieved, before the administration of the next CMS dose (pre-dose or trough). In fact, when the steady state has been achieved, the concentration of colistin at the different times of the dosing interval are quite similar because the plasma curve is almost a flat line (Couet, et al. Clin, Microbiol Infect, 2012).
Point 2: Authors suggested the safe use of colistin in patients with CKD. However, how clinicians adjusted dosage of CMS according to renal function has to be shown in order to ensure the safety. Authors should describe the algorism of dosage adjustment in predefined study protocol in detail.
Response 2: According to the reviewer’s suggestion, the dosage CMS algorithm for all patients based on their baseline GFR has been added to the manuscript (pag 9, line 263-272).
Point 3: Authors should create additional (supplementary) table showing the results of multivariate analysis described in Line 102.
Response 3: As recommended, a new table (Table 5) showing the results of the multivariate analysis has been included in the manuscript (page 6).
Point 4: Based on this study, can authors suggest the therapeutic target Css level of 1mg/L three to four hours after administration is appropriate in patients with CKD? The reviewer seems that this is an impact finding in this study.
Response 4: This is a crucial point of our research on colistin PK/PD. However, the design of this study, does not allow to draw conclusions in this regard. As clarified in the manuscript (pag 7, lines 192-198), these targets could be different for the different infectious diseases and new studies are needed to asses this interesting issue.
Minor comments
Point 5: In Table1, legend is needed to explain abbreviations and median or mean values.
Response 5: As recommended, a legend with different abbreviations used has been added to the tables
Point 6: In study protocol, were patients with eGFR<10 or ESRD excluded?
Response 6: No, patients with eFGR < ml/min or ESDR were not excluded because this not was an exclusion criterion but no patients with these characteristics were included during the study period. This point has been clarified in the Result section (pag 2, line 76-77)
Point 7: In methods, authors should describe the approval number assigned by ethical committee.
Response 7: This information has been added (pag 8, line 236)
Point 8: Did all patients receive combined antibiotic treatment including colistin? If not, percentage of combined treatment should be shown in Table 1.
Response 8: As suggested by the reviewer, this information has been added to Table 1
Round 2
Reviewer 2 Report
The reviewer has no more concerns in the revised manuscript.
Author Response
Dear reviewer:
Thank you very much for your kindly revision of our manuscript.
Luisa
This manuscript is a resubmission of an earlier submission. The following is a list of the peer review reports and author responses from that submission.